# The Potential Role of PPARs in the Fetal Origins of Adult Disease

**DOI:** 10.3390/cells11213474

**Published:** 2022-11-02

**Authors:** Jun Guo, Jue Wu, Qinyuan He, Mengshu Zhang, Hong Li, Yanping Liu

**Affiliations:** 1Center of Reproduction and Genetics, The Affiliated Suzhou Hospital of Nanjing Medical University, Suzhou Municipal Hospital, Gusu School, Nanjing Medical University, Suzhou 215006, China; 2Institute for Fetology, The First Affiliated Hospital of Soochow University, Suzhou 215006, China; 3Department of Obstetrics and Gynecology, Suzhou Dushu Lake Hospital, Clinical College of Soochow University, Suzhou 215006, China; 4Department of Obstetrics & Gynecology, The Second Hospital of Nanjing, Nanjing University of Chinese Medicine, Nanjing 210000, China

**Keywords:** fetal origins of adult disease, PPARs, early development, metabolic, epigenetic

## Abstract

The fetal origins of adult disease (FOAD) hypothesis holds that events during early development have a profound impact on one’s risk for the development of future adult disease. Studies from humans and animals have demonstrated that many diseases can begin in childhood and are caused by a variety of early life traumas, including maternal malnutrition, maternal disease conditions, lifestyle changes, exposure to toxins/chemicals, improper medication during pregnancy, and so on. Recently, the roles of Peroxisome proliferator-activated receptors (PPARs) in FOAD have been increasingly appreciated due to their wide variety of biological actions. PPARs are members of the nuclear hormone receptor subfamily, consisting of three distinct subtypes: PPARα, β/δ, and γ, highly expressed in the reproductive tissues. By controlling the maturation of the oocyte, ovulation, implantation of the embryo, development of the placenta, and male fertility, the PPARs play a crucial role in the transition from embryo to fetus in developing mammals. Exposure to adverse events in early life exerts a profound influence on the methylation pattern of PPARs in offspring organs, which can affect development and health throughout the life course, and even across generations. In this review, we summarize the latest research on PPARs in the area of FOAD, highlight the important role of PPARs in FOAD, and provide a potential strategy for early prevention of FOAD.

## 1. Introduction

In 1989, David Barker and his colleagues performed an epidemiological survey. They found that both newborn deaths and the increased risk of death from stroke and coronary heart disease in adults were related to low birth weight [1]. Later studies have confirmed that low birth weight is linked to a variety of chronic disorders, such as hypertension [2], type 2 diabetes (T2DM) [3], autoimmune thyroid disease [4], and chronic bronchitis [5]. This led to the fetal origin of adult diseases (FOAD) hypothesis that the roots of adult metabolic and cardiovascular disorders lay in the effects of malnutrition in fetal life and early infancy [6].

The FOAD hypothesis builds on the “developmental plasticity” that the organisms exhibit plastic or sensitivity in response to environmental influence during critical developmental periods to improve the match between phenotype and environment [7]. For example, a fetus will undergo the process of remodeling and altering the structure or function of various organs, which is critical for survival as well as neurodevelopment when confronted with the adversity of malnutrition [8]. However, it is important to recognize that a person’s response to environmental stimulation or pathological conditions can be limited, and such an evolutionary advantage of “plasticity” is lost over time [9]. This phenomenon called “programming” shows how early-life stimuli may lead to lifelong and irreversible changes [10]. The FOAD hypothesis attracted a lot of attention in the field of developmental plasticity.

With the expansion and deep-going of research, the recognition that “programming” occurs not only during the fetal period but also during the whole process of life development, including the early embryonic period, infancy, and early childhood [11]. FOAD hypothesis has been expanded and recognized as the Developmental Origins of Health and Diseases (DOHaD). The DOHaD theory states that the interplay between genes and environments (nutrition, stress, or environmental chemicals) from fertilization to the neonatal stage affects the disease risks related to lifestyle in later periods of life [12]. Research regarding the potential mechanisms of adverse stimuli in utero or early stage of life increases the risk of diseases later in life has been a focus of the various current animal and clinical studies [13]. One of the most exciting emerging themes in the DOHaD field is epigenetics [14]. Epigenetic mechanisms typically include DNA methylation, histone modifications, and non-coding RNAs (ncRNAs) [11]. These epigenetic modifications may have long-term consequences for gene expression and may be involved in the occurrence and/or progression of various diseases in postnatal life.

Peroxisome proliferator-activated receptors (PPARs) belong to the nuclear receptor superfamily and perform a broad range of physiological functions, including cellular development, differentiation, energy homeostasis, and metabolism [15]. Numerous studies have shed light on the involvement of the PPARs in multiple system impairments or protective effects against impairment, such as the nervous system, cardiovascular system, and metabolism system [16].

To date, three members of PPARs (PPARα, PPARβ/δ, and PPARγ) have been identified [17]. These nuclear receptors play important roles in cell differentiation, development, and reproduction [18]. All of the PPAR isoforms are identified in the rat ovary [19]. PPARα and PPARβ/δ are present primarily in the theca and stroma. PPARγ is localized mainly in the granulosa cells surrounding and supporting the oocyte meiotic maturation [20]. PPARγ expression increases from the primary/secondary follicle stage to the large follicle stage [21]. However, one study found that its expression level remains consistent during follicle development [20]. The absence of PPARα has no discernible impact on the fertility of mice, whereas the deletion of PPARγ and PPARβ/δ does [22]. Chaffin et al. held the opposite opinion; activation of PPARγ appears to play an inhibitory role in follicular growth and differentiation, according to their research [23]. The three PPAR isoforms are expressed in both somatic and germ cells in the testis [24]. Although the action of PPARs in testis development is still unclear [25], several published studies suggest that male fertility may be influenced by PPARs-regulated lipid metabolism, particularly the β-oxidation of fatty acids [25].

The PPARs isotypes are expressed in the placenta and play an important role in modulating embryo implantation and placental development [26]. Mutation of PPARβ/δ drastically influences placenta development and even embryonic death [27]. PPARγ is necessary for the formation of the labyrinthine layer of the placenta. Mice with deletion of the PPARγ gene exhibit defective placental vascular discourse and embryonic lethality [28]. During fetal development, the interaction of PPARα and its ligands in the liver may be important for the nutrient supply that the fetus may encounter after birth [29]. In addition, there is strong evidence that PPARβ/δ and PPARγ regulate the expression of genes involved in sarcoplasmic and adipose tissue production [30]. Remarkably, PPARs play a crucial role in metabolism during early life, and alterations in PPAR metabolic pathways could be one candidate mechanism contributing to the FOAD [31]. The recent work in developmental epigenetics has significantly expanded our understanding of this interaction. Exposure to adverse events in early life can affect the methylation pattern of PPARs in multiple organs, such as the brain, lungs, heart, blood vessels, liver, and skeletal muscles. Various chronic adult diseases, especially diabetes, cardiovascular disease, and chronic lung disease, show a clear association with PPARs (Figure 1). This review summarizes the contributions of PPARs to the potential mechanisms involved in the FOAD in order to provide a new theoretical direction for the early prevention and even treatment of fetal origin diseases.

## 2. Early Life Adverse Exposure and Future Disease Risk

Epidemiological studies in humans have demonstrated an association between the quality of the early life environment and future disease risk [32]. Animal studies provided insight into the potential mechanisms for these observations by highlighting the environmentally induced changes to epigenetic marks during development [33]. The FOAD theory goes beyond nutrition assumptions and links fetal development to many other exposure factors, such as obesity [34], prenatal maternal stress [35], and environment [36]. Notably, a growing body of research indicates that the paternal environment and dietary habits influence disease onset in offspring [37].

### 2.1. Nutrition

The investigations have shown a strong correlation between maternal food and nutritional status and fetal development and child health [38]. Maternal diet has often been shown to affect subsequent phenotypic [39]. Many diseases such as type 2 diabetes [40], cardiovascular disease [41], and certain cancers [42] are related to low birth weight. Early epidemiological research used data from several well-documented famines and historical cohorts [43]. The most well-known is the Dutch famine cohort. Studies on the Dutch Hunger Winter have provided convincing evidence. Prenatal exposure to famine, especially in the third trimester, has been reported to be associated with decreased glucose tolerance in adults [44]. Even if the effect on fetal growth is minimal, malnutrition in utero may result in long-term alterations in insulin-glucose metabolism [44]. Additionally, individuals whose mothers were exposed to the Dutch Famine before or during gestation were almost three times more likely to develop hypertension in adulthood than unexposed adults [43]. A study of 290 men born in East Hertfordshire during 1920–1930 showed that the risk of coronary heart disease is increased in children with low birth weight [45]. Apart from the above, the studies of the 1959–1961 mass famine in China and the 1944–1945 Dutch Hunger Winter have found a link between poor nutrition early in life and mental health and cognitive development [46].

Early research focused on evidence linking maternal protein restriction or malnutrition to the long-term health of offspring. Nowadays, a fat-rich diet is prevalent around the world, and 50% of women of childbearing age are overweight or obese in the US [47]. The impact of high-energy-dense, high-glucose, and high-salt foods during pregnancy on the phenotype of the offspring is being studied using animal models. Offspring whose mothers received these various diets showed persistent metabolic changes that were comparable to human cardio-metabolic disorders such as hypertension, insulin resistance, and obesity [48].

Obesity is also highly prevalent among adult men. The prevalence of overweight or obesity (BMI ≥ 25) is 72.1% in men and 61.2% in women, according to recent national statistics on the US population [49]. In 2000, Figueroa et al. published the first research revealing the parental influence on child health in humans [50]. They found that fathers’ total and percentage body fat were predictors of changes in body fat of premenarcheal girls during a 2.7-y period [50]. In 2010, Ng et al. reported that a high-fat diet in male rats resulted in β-cell dysfunction in F1 female offspring [51].

### 2.2. Environment

In modern society, humans are exposed to a wide range of environmental chemicals, such as endocrine disruptors and other toxins from lifestyle habits [52]. Numerous epidemiological studies have shown that prenatal exposure to multiple environmental pollutants has an impact on fetal development [53]. There have been many investigations that found a connection between four major environmental pollutants (perfluorinated compounds, polyhalogenated aromatic hydrocarbons, heavy metals, and air pollutants) and impaired fetal development and lower birth weight in humans [54]. A study of 1277 children from the European HELIX (Human Early Life Exposure Group) cohort reveals that BP in children may be influenced by early exposure to some substances, as well as the built environment and climatic conditions [55]. Miguel et al. summarize the influence of the early environment on the structural and functional development of children’s brains in their review [56]. For instance, most drugs of abuse (e.g., ecstasy, opiates) can readily cross the placenta and impact fetal brain development [57]. The genes associated with brain growth, myelination, and neuronal migration were down-regulated in the brain of a fetus exposed to tobacco in utero [58]. Several large population-based cohort studies have shown that prenatal exposure to maternal smoking during pregnancy or smoking cessation in early pregnancy was significantly associated with childhood ≈ [59]. Prenatal ethanol exposure (PE) impairs dopaminergic (DA) neuron function in the midbrain [60]. Air pollution can affect the anatomy and physiology of the umbilical cord and placenta [61]. Particles induce antiangiogenesis, resulting in the thinner and less voluminous umbilical cord in mouse models, which affects oxygen transport [62] and replicates in humans [63]. A meta-analysis of epidemiological studies suggests that exposure to air pollution increases the risk of pregnancy-induced hypertensive disorders [64].

A cross-sectional study of 67 men in North Carolina indicated that exposure to environmental chemicals/factors (organophosphates) could alter DNA methylation in human sperm cells, thereby affecting the health of offspring [65]. It has also been shown that human exposure to bisphenol A affects the global methylation of sperm DNA [66]. Environmental toxins also include lifestyle habits such as smoking and alcohol intake. Chronic consumption of smoking and alcohol was associated with epigenetic abnormalities and altered miRNA expression in spermatozoa [67].

### 2.3. Stress

In addition to physical status, the effects of altered maternal mental health and psychological stress during pregnancy on the offspring have been extensively documented in the literature. Maternal anxiety, depression, and stress disorders are common in pregnant women. A wide range of acute and chronic maternal stress exposures, such as daily hassles [68], life event stress [69], and unusual and extremely stressful events [70], have a negative impact on child development [71]. Low birth weight in infants is linked to chronic maternal stress, racism exposure, and depressive symptoms during pregnancy [72]. Accumulating research indicates that prenatal stress and depression during pregnancy are associated with cognitive and academic performance difficulties [73]. Maternal anxiety during pregnancy is associated with subsequent infant development, increased risk of behavioral/emotional disorders, and depression later in children [74]. According to electroencephalography and MRI research results, infants whose mothers had prenatal anxiety may have less volume and/or thickness in their frontal, temporal, and limbic regions and more frontal activity [75]. Natural changes in maternal care during the first day of life are associated with long-term changes in stress reactivity and hippocampal morphology and function in rodent studies [76]. These effects are mediated by epigenetic changes in the promoter of the progeny hippocampal glucocorticoid receptor gene [77]. In humans, childhood abuse was similarly associated with increased DNA methylation and decreased hippocampal glucocorticoid receptor expression [78]. Prenatal stress exposure has been linked to neurodevelopment and the risk of neuropsychiatric disorders in offspring [79]. Retrospective epidemiological studies have provided compelling evidence linking lifetime stress exposure in men with disease risk in offspring [80]. The rodent studies have demonstrated the susceptibility of germ cells to stressful environments throughout the paternal lifetime [81].

## 3. Peroxisome Proliferator-Activated Receptors

Peroxisome proliferator-activated receptors (PPARs) belong to nuclear hormone receptors (NRs) and ligand-activated transcription factors that regulate genes crucial for cell differentiation and a variety of metabolic processes such as glucose and lipid homeostasis. The PPAR family consists of three different isoforms: PPARα, PPARβ/δ, and PPARγ. These three isotypes have different tissue distribution, biological activity, and affinity for ligands [82]. The essential roles of PPARs in regulating mitochondrial function and energy metabolism have been clearly established. Notably, all three PPAR subtypes have overlapping and also distinct functions in regulating metabolic processes. Six functional domains (from A to F) make up the PPARs [83]. A C structural domain is present at the N-terminus of PPAR, also known as the DNA binding domain (DBD). The DNA sequence in the promoter region of genes, called the peroxisome proliferator response element (PPRE), is recognized by DBD. On the other hand, a ligand-binding domain (LBD) in the C-terminus is responsible for the specificity of the ligand and dimerization of the receptor with the retinoid X receptors (RXR) [84]. PPARs translocate to the nucleus after interacting with specific ligands (synthetic or non-synthetic) [85]. PPARs interact with RXR, peroxisome proliferator-activated receptor gamma-coactivators (PGC), steroid receptor coactivators, and CREB binding protein (CBP/p300) after translocating to the nucleus, then bind to the sequences of PPRE, which subsequently initiate the transcription of target genes involved in different physiological processes [86]. The target genes are involved primarily in the metabolism of fat, as well as in cellular proliferation and differentiation, protein and glucose, inflammation, and tumorigenesis [86]. Their aberrant expression is related to a variety of disorders, such as neurodegenerative disorders, cardiovascular disease, obesity, type 2 diabetes, pancreatic cancer, and so on [16].

Given their central roles in regulating metabolic flexibility, it is essential to understand the manner in which PPARs regulate gene expression. The function of PPARs is principally modulated by ligand binding, which induces structural changes, further recruiting co-activator or co-repressor complexes, which stimulate or inhibit their functions [87]. In addition to ligand binding, post-translational modifications of PPARs are emerging as one such way PPARs are regulated, including phosphorylation, ubiquitination, SUMOylation, acetylation, and O-GlcNAcylation, which contribute to fine-tuning of the transcriptional activities [88]. Recent studies have suggested that post-translational modifications are observed in all three PPAR isoforms [87,88]. A detailed view of the functional regulation of PPARs through post-translational modifications can be found in a recent review by Xu et al. [87].

### 3.1. PPARα

PPARα is known to be important for regulating the transcriptional expression of key enzymes that are involved in mitochondrial dynamics and metabolic functions, including glucose metabolism, fatty acids β-oxidation, and fatty acid transport [89]. Moreover, PPARα receptors are found largely in metabolically active tissues, such as brown adipose, skeletal muscle, heart, liver, and intestinal mucosa tissues. Natural ligands for the PPAR receptor include saturated, monounsaturated, and polyunsaturated fatty acids and their metabolites, such as leukotrienes B4, oxidized phospholipids, lipolytic lipoprotein products, etc. Nature ligands bind to PPARα and activate PPAR-responsive genes, increasing hepatic intracellular fatty acid absorption [90]. PPARα also plays an important role in extracellular lipid homeostasis by modulating the transcriptional regulation of major very-low-density and high-density apolipoproteins [91]. Furthermore, PPARα seems to modulate the bioactivity of leptin in the liver and adipose tissue [92].

The transcriptional activity of PPARα is enhanced by binding to the ligands, after which transcriptional coactivators contribute to the activation of target genes [93]. In addition, PPARα trans-activity is regulated by post-translational modifications such as phosphorylation, SUMOylation, and ubiquitination. As a phosphoprotein, PPARα is phosphorylated exclusively on serine residues in vivo [94]. It was reported that treatment with insulin or ciprofibrate (a PPARα agonist) increased the phosphorylation of PPARα [95,96]. SUMOylation is a reversible post-translational modification that has been established as one of the key regulatory protein modifications in eukaryotic cells. Two lysine residues of PPARα, K185, and K358, have been reported to be modified by SUMOylation [97,98]. Moreover, several studies have shown that the ubiquitin–proteasome system is involved in the regulation of PPARα activity. These studies suggest the ubiquitination of PPARα in a ligand-dependent manner, and that effect of ubiquitination on PPARα activity depends on the systems studied [87,88].

### 3.2. PPARβ/δ

PPARβ/δ is generally expressed in nearly all tissues, such as the brain, skin, liver, skeletal muscle, heart, and various types of cancer [99]. Polyunsaturated fatty acids (arachidonic and linoleic acids) and their metabolites, such as prostacyclin PGI2, are suggested as natural ligands. Similar to the other PPAR family members, it mainly participates in the oxidation of fatty acids and affects lipid metabolism, both reducing fat and hence preventing the development of obesity and controlling blood sugar and cholesterol levels in the heart and skeletal muscle [100]. Overall, PPARβ/δ has significant functional overlap with PPARα in most tissues. For example, PPARβ/δ also stimulates fatty acid oxidation in muscle and heart [101]. However, PPARβ/δ and PPARα carry out different roles in regulating hepatic energy metabolism. Unlike PPARα, PPARβ/δ regulates gene expression associated with lipogenesis and glucose utilization rather than inducing fatty acid oxidation [102]. Additionally, several investigations have shown a large expression of PPARβ/δ in the central nervous system (CNS) [103]. PPARβ/δ may affect the differentiation of neural and glial cells and alter cholesterol metabolism in the brain. It is well known that PPARs regulate inflammatory processes associated with lipid signaling pathways [104]. Inhibiting inflammatory processes in the CNS may reduce brain damage and enhance motor and cognitive outcomes [105]. Comparatively, PPARβ/δ is the least reported PPAR family member in terms of a post-translational modification. To date, we are aware of only one previous study showing that SUMOization of PPARβ/δ at K104 is removed by SENP2 and promotes FAO gene expression in muscle [106].

### 3.3. PPARγ

PPARγ is widely expressed in brown and white adipose tissue, spleen, and large intestine. PPARγ has two isoforms in mice and four different isoforms in humans [107]. Unsaturated fatty acids and their metabolites are the primary natural modulators of PPARγ. Activated PPARγ by these natural ligands regulates adipogenesis and fat distribution, the levels of adipokines such as adiponectin, which involve insulin sensitivity and lipid and glucose metabolism [108]. PPARγ are ligand-inducible transcription factors involved in regulating the expression of genes, including glucose sensitivity (IRS-1, IRS-2, GLUT-4, and PI3K), fatty acid uptake and mobilization (FAT/CD36, FABPs, and LPL) and triglyceride synthesis (ACSL, GK, and PEPCK) [102]. In addition, PPARγ also induces the expression of mitochondrial proteins, such as CPT-1 and UCPs, which play an important role in the regulation of mitochondrial metabolism. PPARγ is associated with the pathology of many diseases, such as obesity, atherosclerosis, diabetes, and cancer. PPARγ agonists, including troglitazone, rosiglitazone, ciglitazone, and pioglitazone, have been used in the treatment of hyperlipidemia and hyperglycemia [109]. The role of PPARγ in cancer initiation/progression is contradictory. Numerous studies show that PPARγ has a tumor-promoting effect. Conversely, some literature has reported that PPARγ plays a key role in tumorigenesis as a tumor suppressor. PPARγ activation by many agonists has been demonstrated to have antiproliferative and proapoptotic actions in prostate, thyroid, and lung cancers [110].

Thiazolidinediones, such as rosiglitazone, pioglitazone, and lobeglitazone, are PPARγ agonists that modulate the transcriptional activity of PPARγ. Like PPARα, PPARγ activity is also regulated by post-translational modifications. PPARγ is now known to be phosphorylated upon stimulation of the MAPK activation pathway [87]. A variety of stimuli (growth factors, platelet-derived growth factors, transforming growth factor beta, insulin, and prostaglandin F2 alpha, etc.) can activate PPARγ phosphorylation via specific activation of MAPKs [87]. Moreover, PPARγ is regulated by SUMO1 and SUMO2 sumoylation. The targeted lysine residue was identified as K107, K33, K64, K68, and K77, respectively [111,112]. In addition to this, other post-translational modifications of PPARγ, such as acetylation, ubiquitination, and O-GlcNAcylation, have also been reviewed by Xu et al. [87].

## 4. Effects of PPARs in the Placenta and the Fetus

During pregnancy, the placental metabolism can adapt to the environment throughout pregnancy to adapt to the maternal nutritional status and meet the demands of the fetus [31]. All three PPAR isoforms are expressed in the placenta [26,113]. The PPARs promote placental developmental plasticity by regulating lipid, hormone, and glucose metabolic pathways, including lipidogenesis, steroidogenesis, glucose transporters, and placental signaling pathways [114]. Although the role of each PPAR in placental function has not been fully determined, unique and common functions between these isoforms have been observed. Among the PPAR-isoforms, PPARγ appears to be a major regulator of the mammalian placenta [115]. PPARγ was the first to be demonstrated in the placenta [116]. In rodent placenta, PPARγ is largely expressed in the trophoblastic layer of the labyrinth zone [117,118]. In human placenta, PPARγ is present in villous trophoblast and extravillous trophoblast [119,120]. There is some evidence suggesting that PPARγ modulates villous trophoblast differentiation, oxidative pathways, inflammatory response, and barrier formation [121,122]. Furthermore, dysregulation of both PPARα and PPARγ in the placenta has been implicated in common complications of pregnancy, such as gestational diabetes mellitus, intrauterine growth restriction, and preeclampsia [123]. Their expression pattern is regulated at least partially by DNA methylation in the placenta, and the involvement of other PPAR-regulated processes, such as placenta-specific miRNAs, has just been discovered [124]. Placental epigenetic regulation appears to provide a plausible connection between environmental exposures and fetal development. Studies have shown that changes in placental DNA methylation patterns have been associated with fetal growth after exposure to maternal risk conditions such as GDM, obesity, and preeclampsia [125,126].

During the development of the human embryo and fetus, three isoforms are expressed in cells of the endoderm and mesoderm at early time points in gestation [127]. PPARβ/δ was the first allotype to be expressed during embryonic development in rodents [128]. PPARα and PPARγ are expressed first in the placenta and then in the fetus [128,129]. The role of PPAR in development has been revealed by studies in PPAR knockout mice [130]. The important role of PPARα in lipid catabolism in the fetal liver and heart is consistent with the function of PPARα in adult tissues [131,132,133]. Knockout of PPARα in mice causes a high miscarriage rate, hepatic lipid accumulation, obesity, and prolonged inflammation [134,135]. In the early stages of organogenesis in the rat embryo, only the PPARβ/δ isotype is expressed [128] and plays an important role in the closure of the neural tube [136]. The signaling pathway involved in PPARβ/δ activation associated with nervous system development is profoundly altered by maternal diabetes [136]. PPARγ null mutations are lethal. The developmental defects in the placenta occur in parallel to developmental defects in the embryo [137]. In fetuses from diabetic rats, the concentration of PPARγ endogenous is reduced [138]. The capacity of PPARγ endogenous to prevent the overproduction of both NO and MMPs in fetuses from diabetic rats demonstrates its anti-inflammatory effects [138]. PPARγ activation increases lipid concentrations in midgestation fetuses from diabetic rats [139]. Collectively, these data indicate that PPARs-mediated mechanisms are involved in the fetal origins of health and disease.

## 5. PPARs and FOAD

It is now well recognized that adverse events exposure in early life contribute to the development of the chronic diseases of adulthood, including hypertension, type 2 diabetes, stroke, cognitive impairment, and pulmonary hypertension. Additionally, the role of PPARs in numerous chronic diseases such as diabetes, cardiovascular diseases, autoimmune diseases, chronic fatigue, depression, and neurodegenerative diseases is well established. PPARs are ubiquitously expressed in almost all mammalian tissues and organs. Altering PPARs methylation patterns during early development may be maintained throughout the life course and even across generations [31]. In the following sections, we review the expression pattern of PPARs in various organs, including the brain, lung, heart, vessel, liver, and skeletal muscle, and discuss the potential roles of PPARs in FOAD (Table 1).

### 5.1. Brain

PPARα is expressed in several regions of the central nervous system (CNS), and its specific biological function remains unclear [171]. Various inflammatory parameters were significantly enhanced in PPARα KO mice [172]. Neuroinflammation is considered a cause and/or contributing factor to neuronal degeneration [173]. It suggests that PPARα attenuates the inflammatory response after ischemia/brain injury [174]. Moreover, the activation of PPARα has anti-inflammatory properties and a beneficial impact on certain neurologic diseases, including Alzheimer’s disease (AD) [175], Multiple sclerosis (MS) [176], Huntington’s disease (HD) [177], and Parkinson’s disease (PD) [178]. Malnutrition during pregnancy affects sleep homeostasis and increases sleep pressure in offspring, which may be related to the increased PPARα mRNA expression in the hypothalamus [140]. In a study by Felice et al., it was found that prenatal administration of fenofibrate (PPARα agonist) reduced the risk of schizophrenia-like behavior in male offspring of maternal immune activation (MIA) and emphasizes PPARα as a possible target for schizophrenia therapies [141].

Although PPARβ/δ is the most abundant PPAR subtype in the CNS, its role is rarely studied [171]. It has been suggested that the most important roles of PPARβ/δ in brain cells are antioxidant and anti-inflammatory effects [179]. There also a study identified that the differentiation of neural and glial cells might be impacted by PPARβ/δ, which may also affect the metabolism of cholesterol in the brain [103]. One study found that prenatal exposure to a high-fat diet increased the density of cells immunoreactive for PPARβ/δ in the hypothalamic paraventricular nucleus, perifornical lateral hypothalamus, and central nucleus of the amygdala [180]. However, the clinical significance of this change and the potential role of PPARβ/δ in fetal origins of CNS diseases remains unclear.

PPARγ is the most studied subgroup of the PPAR family and has an important role in the CNS, including relieving endoplasmic reticulum stress and the inflammatory response [181], the balance of cerebral metabolite [182] and the maintenance of glucose homeostasis [183]. Animal studies have demonstrated that maternal vitamin D deficiency leads to decreased PPARγ levels in the offspring’s brain and affects angiogenesis in the brain [142]. Fetal hippocampal inflammation is significantly increased in immune-activated mothers, followed by cognitive deficits, which are highly correlated with hippocampal neurogenesis disorders in pre-pubertal male offspring. The PPARγ agonist pioglitazone improves neurogenesis, cognitive impairment, and anxious behavior in MIA offspring [143]. Maternal high-fructose-induced hippocampal neuroinflammation in the adult female offspring. Adult female offspring exposed to high maternal fructose have decreased levels of PPARγ and endogenous antioxidant expression in the hippocampus, which leads to hippocampal neuroinflammation. An oral dose of pioglitazone (PPARγ agonist) effectively increases the expression of antioxidants and blocks neuroinflammation [144]. Based on the findings described above, synthetic PPARγ agonists have been suggested as therapeutic medicines for the treatment of CNS diseases such as PD [184], AD [185], HD, and Autism spectrum disorder [186].

### 5.2. Lung

PPARα has been implicated in the control of airway inflammation, but as yet, little is known about its role in lung disease. There is a mouse model of pulmonary fibrosis suggesting that PPARα regulates fibrosis [187]. A study by Genovese et al. revealed that endogenous and exogenous PPARα ligands reduced bleomycin-induced lung injury in mice [188]. Liu et al. found that the activity of PPARα was inhibited in lipopolysaccharide (LPS) induced acute lung injury (ALI) [189]. By reducing oxidative stress and inflammation, which are both directly related to the activation of PPARα, eupatilin has a protective function in ALI [190]. Taken together, they proposed that PPARα could be a potential therapeutic target for lung injury.

PPARβ/δ agonist inhibited the proliferation of lung fibroblasts and enhanced the antifibrotic properties of PPARγ agonist [187]. The role of PPARβ/δ in pulmonary hypertension and lung cancer has received attention in recent years. According to epidemiological and experimental animal studies, prenatal hypoxia, intrauterine growth restriction (IUGR), and obesity raise the risk of pulmonary hypertension in offspring [191]. Prostacyclin and prostacyclin mimetics are the cornerstone of treatment for patients with pulmonary arterial hypertension (PAH) [192]. One study suggests that PPARβ/δ may be a potent target for prostacyclin mimics in the treatment of pulmonary hypertension. PPARβ/δ agonist (GW0742) mediates vascular relaxation and prevents the right heart from hypertrophy associated with pulmonary arterial hypertension [193]. The role of PPARβ/δ in the negative growth regulation of lung cancer cells was first reported in an in vitro study [194]. Using a variety of lung cancer models, one research group demonstrated that increased synthesis of the PPARβ/δ agonist (prostacyclin) inhibited lung tumorigenesis [195]. These findings imply that PPARβ/δ may play a protective function in PAH and lung cancer.

PPARγ is expressed in many lung cells, including bronchial epithelial cells, airway smooth muscle (HASM) cells, fibroblasts, alveolar type II pneumocytes, and mononuclear phagocytes [187,196]. The activation of PPARγ signaling is involved in the paracrine effect of interstitial fibroblasts and alveolar type II (ATII) cells, which is necessary to maintain alveolar homeostasis [197]. The PPARγ gene depends on developmentally specific transcription of mRNA variants and epigenetics for normal tissue. Therefore, it is susceptible to epigenetic changes [198]. There is evidence that perinatal damage, including exposure to nicotine or maternal tobacco smoke (MTS), IUGR, and preterm delivery, altered both epigenetic determinants and gene expression in the lung [198]. It has been demonstrated that IUGR caused epigenetic modifications to the PPARγ gene in rat lungs [199]. The levels of PPARγ mRNA variants, PPARγ protein, and downstream targets were decreased in the lung of neonatal rats [149]. Numerous studies have shown an increase in asthma in offspring whose mothers smoked during pregnancy [200]. Perinatal smoke/nicotine exposure is a recognized factor that affects lung growth and differentiation by down-regulating the expression of PPARγ [201]. Downregulation of PPARγ causes lipid-rich alveolar mesenchymal fibroblasts to transdifferentiate into myofibroblasts, which is the cellular hallmark of chronic lung diseases such as asthma [202,203]. PPARγ agonist (rosiglitazone) can effectively block asthma induced by perinatal smoke exposure [148].

### 5.3. Heart

PPARs are the physiological master switches of the heart, which guide the energy metabolism of cardiomyocytes, thereby influencing pathological heart failure and diabetic cardiomyopathy [204]. However, the roles of PPARs in heart function and the results of their respective agonists differ greatly in preclinical animal models and clinical studies [205]. PPARα is highly expressed in the heart and can affect the expression of numerous genes implicated in the uptake and oxidation of cellular fatty acid (FA) [206]. Therefore, it plays a major role in cardiac fatty acid homeostasis. Down-regulation of PPARα expression altered the expression of fatty acid-metabolizing proteins that lead to myocardial damage and fibrosis [207]. The expression of fetal cardiac lipid metabolism genes (PPARα, fatty acid translocase, lipoprotein lipase, etc.) was reduced in offspring from mothers with high blood glucose levels, not accompanied by cardiac triglyceride deposition or cardiac hypertrophy [132]. However, it has subsequently been suggested that the heart of adult offspring from diabetic rats showed increased lipid concentrations. The increased expression of PPARα in offspring from diabetic rats can prevent toxic lipid accumulation in the heart [208]. There is also solid evidence that PPARα exerts a protective effect on cellular oxidative damage [209]. Thus, chronic deactivation of the PPARα signaling pathway may disrupt the balance between oxidant production and antioxidant defenses and ultimately contribute to heart damage [210]. In the 2-day-old and pre-pubertal stage progeny from diabetic rats, there was an increase in the expression of prooxidative/proinflammatory markers and PPARα protein expression in the hearts. Maternal treatment with mitochondrial antioxidants led to reductions in PPARα protein expression and pro-oxidant/ro-inflammatory markers and prevented the adverse programming of heart alterations in prepubertal offspring from diabetic rats [151]. Both neonatal and adult hearts from the offspring of maternal protein restriction (PR) during pregnancy showed a reduction in the level of PPARα promoter methylation and an increase in PPARα mRNA expression [150]. The possible implication of these findings is that the enhanced capacity of fatty acid β-oxidation leads to an increased risk of oxidative damage to offspring hearts.

PPARγ is expressed at very low levels in the adult heart [211]. PPARγ activation in cardiomyocytes is associated with impaired cardiac function due to its lipogenic effect [211]. Maternal obesity leads to cardiac hypertrophy, and left ventricular diastolic dysfunction in offspring might be related to persistent upregulation of PPARγ expression [152]. In rat offspring programmed by the reduced protein diet during gestation, the PPARγ agonist (rosiglitazone) was shown to have beneficial effects by reducing cardiac fibrosis and enhancing myocardial vascularization [153]. PPARγ activator therapy has a beneficial impact on risk factors for cardiovascular disease, but it also appears to have adverse effects on the cardiovascular system. It has been reported that treatment with rosiglitazone is associated with an increase in myocardial infarction (MI) or heart failure in humans [212].

### 5.4. Vessel

Studies have shown that PPARs are present in all essential vascular cells, including monocyte-macrophages, endothelial cells, and vascular smooth muscle cells [213]. PPARs influence lipid metabolism and vascular diseases such as atherosclerosis and hypertension [214]. PPARα has been implicated in blood pressure regulation and vascular inflammation [215]. PPARα was expressed in both vascular endothelial cells and vascular smooth muscle cells [216]. Activation of PPARα blocks multiple pathways such as NF-κB and MAPK, which in turn inhibit the expression of many genes involved in vascular inflammation, oxidative stress, and cell growth and migration [217]. In experimental hypertension models, PPARα ligands can reportedly lower blood pressure [218]. PPARα was also associated with atherosclerotic processes [219]. The administration of the fibrate class of PPARα agonists to patients with type 2 diabetes or dyslipidemia significantly slowed the development of atherosclerosis and reduced their risk of cardiovascular events [220], but surprisingly, high-fat diet PPARα-null mice are more responsive to insulin, have lower blood pressures, and develop less atherosclerosis [219].

Activation of PPARβ/δ has a significant effect on anti-hypertension [221]. However, it is argued that PPARβ/δ agonist acts via interference with the ET-1 signaling and lower blood pressure through a PPARβ/δ-independent mechanism [222]. Moreover, the reduction of vascular oxidative stress markers and improvement of endothelial dysfunction were observed after a high dose of the PPARβ/δ antagonist GSK0660 [223]. It has been shown that the offspring of rats with maternal diabetes have abnormal fetal programming of vascular endothelial function, which is linked to increased ER stress and may be attributed to the down-regulation of the AMPK/PPAR signaling cascade [224].

Whether PPARγ is hypotensive or hypertensive is still under debate so far [225]. Genetic studies showed impaired vascular smooth muscle contraction in response to alpha-adrenergic drugs and hypotension in a generalized PPARγ knockout mouse model [226], which is very well in agreement with the findings by Tontonoz [227]. These findings suggest that PPARγ has a hypertensive function in controlling blood pressure. However, activation of PPARγ has beneficial effects on hypertension in a number of animal and human studies [228]. PPARγ activation may regulate blood pressure via modulating endothelial vasoactive factors such as prostacyclin, nitric oxide, and endothelin-1. Additionally, PPARγ may also be involved in vessel tone regulation by down-regulating ANG II receptor 1 (AT1-R) in vascular smooth muscle cells [229]. Angiotensin II-induced endothelial dysfunction in adult offspring of pregnancy complicated with hypertension is associated with impaired endothelial PPARγ [155]. Rosiglitazone (a PPARγ agonist) reduced blood pressure and attenuated vascular remodeling in perinatal low-protein offspring rats [156]. Chronic treatment with rosiglitazone has also been shown to prevent impaired nitric oxide synthase-dependent responses induced by prenatal alcohol exposure [230]. Collectively, it is widely believed that activation of PPARγ can moderately lower blood pressure and plays a protective role in endothelial dysfunction, vascular inflammation, and other pathological processes that lead to atherosclerosis [231].

### 5.5. Liver

The liver is a major organ that regulates whole-body nutrient and energy homeostasis. PPARs are involved in the regulation of adipogenesis, lipid metabolism, inflammation, and metabolic homeostasis [232]. PPARα is a major regulator of lipid metabolism in the liver, especially at fasting. In addition to fatty acid oxidation and ketogenesis, PPARα controls the expression of almost all genes involved in lipid metabolism in the liver [233]. Free fatty acids and other lipids are known to activate PPARα to increase lipid clearance in the liver [234]. In the liver of the PPARα-null mice (lacking the PPARα gene), constitutive mitochondrial β-oxidative activity was significantly reduced [235]. Polyunsaturated fatty Acids (PUFAs) are endogenous PPARα activators. Mice on a high-fat diet supplemented with PUFAs showed enhanced hepatic FA β-oxidation and ameliorated fatty liver [236]. Maternal exposure to perfluorooctanoic acid (PFOA) significantly decreased the expression of the PPARα gene in female offspring mice, leading to reduced fatty acid oxidation and histone acetylation and increased liver oxidative stress [157]. Other authors have found a lower expression of PPARα in the liver of rat offspring exposed to vitamin B12 deficient diets before and during pregnancy due to increased global methylation levels. [237]. The offspring born to an obese mother has a greater likelihood of progression to the fatty liver, which may be associated with PPARα dysfunction [238]. Similar works showed that a high-fat diet during pregnancy impairs the demethylation of PPARα, therefore inducing lipid metabolism disorders and obesity in offspring [161]. Maternal high-fat diet decreased the expression of PPARα and genes for fatty acid oxidation, which contributes to nonalcoholic fatty liver disease (NAFLD) in offspring [158]. Prenatal 1,2-Cyclohexane dicarboxylic acid diisononyl ester (DINCH) plasticizer exposure downregulates PPARα expression, which, in turn, affects the liver function of offspring [239]. Maternal nicotine exposure leads to lipid metabolism disorders and insulin resistance by activating PI3K/Akt signaling, inhibiting PPARα protein expression, and promoting the progression of MAFLD in adult offspring [159].

PPARγ is expressed at much lower levels in the liver and muscle than in adipose tissue and macrophages [240]. Many studies have demonstrated a link between elevated PPARγ expression and hepatic steatosis [241]. Specific disruption of liver PPARγ in mice can effectively improve fatty liver [242]. Overexpression of PPARγ in mouse liver can lead to the development of adipogenic hepatic steatosis [243]. Activation of PPARγ is steatogenic. Paradoxically, treatment of PPARγ-null mice with PPARγ ligands protects other tissues from TAG accumulation and insulin resistance [244]. In A-ZIP/F-1 mice, disrupting hepatocyte PPARγ reduced hepatic steatosis but worsened hyperlipidemia and muscle insulin resistance [244]. PPARγ has anti-inflammatory effects; PPARγ activation decreases inflammatory response by negatively interfering with NF-κB and signal transducers and transcriptional activators [245]. PPARγ agonists may have potential in the prevention of liver fibrosis/cirrhosis [246]. The NAFLD induced by gestational BPA exposure in male offspring may be related to the dysregulation of the HNF1b/PPARγ pathway [165]. Co-agonists of PPARα and PPARγ attenuated liver and white adipose tissue inflammation in male offspring of obese mothers [247]. The reduction of PPARγ level plays a crucial role in arsenic-induced hepatic autophagy in progeny [248]. Metabolic and reproductive disturbances in the female offspring of polycystic ovary syndrome may be associated with the upregulation of PPARγ in the liver [249]. Prenatal exposure to a low-protein diet exhibited a lower expression of PPARγ and hepatic steatosis [250].

### 5.6. Skeletal Muscle

Skeletal muscle is a metabolic organ that accounts for 40% of the total body weight in a healthy person. It produces adenosine triphosphate (ATP) through insulin-mediated glucose uptake, stores excess glucose as glycogen, and is involved in fatty acid oxidation. All three PPAR isotypes have significant effects on muscle homeostasis, either directly or indirectly. PPARα participates in glucose metabolism and fatty acid catabolism, which is crucial in regulating inflammation and energy expenditure [251]. PPARβ/δ is the major PPAR isotype in skeletal muscle. It is involved in lipid and glucose metabolism, energy expenditure, inflammation, tissue repair and regeneration, and muscle fiber type switching associated with physical exercise [252]. One of the main functions of PPARγ in skeletal muscle is fat deposition [253]. Several observations suggest that PPAR isotypes are at least partially related and overlapping in muscle.

Maternal protein restriction impaired the expression of genes that increased the ability to oxidize fat in response to fasting and exhibited an enhanced expression of PPARα in adult offspring [250]. The study by Zhou et al. showed that miR-29a was upregulated in the skeletal muscles of IUGR offspring. The direct interaction between miR-29a and PPARβ/δ inhibited the expression of PPARβ/δ, which was associated with the progression of insulin resistance (IR) [167]. The reduced mitochondrial content in the muscle of IR offspring may be in part due to decreased PPARβ/δ activation [168]. Maternal cafeteria diet during gestation and lactation Maternal cafeteria diets during pregnancy and lactation were associated with the increased PPARγ mRNA level in pups [170]. The adult mice suffered from maternal caloric restriction during late pregnancy, and a post-weaning high-fat diet, the expressions of PPARγ in their skeletal muscle tissue were significantly increased [169]. PPARγ agonist can improve skeletal muscle insulin sensitivity in the pregestational intrauterine growth-restricted rat offspring [254]. In conclusion, skeletal muscle insulin resistance and impaired fat or glucose metabolism may be closely related to PPARs changes in offspring exposed to adverse factors during pregnancy.

## 6. Conclusions and Outlook

The concept of FOAD (or DOHaD) has provided new insights into the origin of lifestyle diseases, and the field of FOAD has grown rapidly to high prominence in biomedical science and public health. This review is concerned with understanding how stressful environmental conditions during sensitive periods of early development influence the risk of chronic disease later in life, particularly the role of PPARs in this process. Notably, agonists of PPARs have been intensively evaluated as a potential strategy for the early prevention of FOAD. A growing body of exciting evidence demonstrates that PPAR activators reverse some of the adverse effects of adverse exposure during pregnancy on offspring. These data provide important proof that the epigenetic state of a particular gene can be modified. It provides a novel therapeutic strategy to prevent or delay the fetal origin of adult diseases through epigenetic regulation of metabolic genes. Here, we briefly summarize the relevant studies in Table 2. Nevertheless, studies on PPARs in the area of FOAD are currently in the nascent stage, especially the application of PPARs agonists in the primary prevention and treatment of FOAD remains controversial. Therefore, further research is necessary to enhance our understanding of the PPAR-mediated mechanisms involved in the fetal origins of health and disease. Connecting early-life adverse events exposures and PPARs epigenomic measures with later-life health outcomes is a proven strategy for investigating such underlying mechanisms. Recent research has begun to identify features of the PPARs-related regulation of non-coding RNAs, histone modification, and DNA methylation in FOAD. These advances drive the development of the complex transcriptional and epigenetic regulation of PPARs in FOAD. We believe that the studies of such new perspectives will open up new avenues in FOAD research, as well as potential strategies for early prevention of FOAD.

## Figures and Tables

**Figure 1 cells-11-03474-f001:**
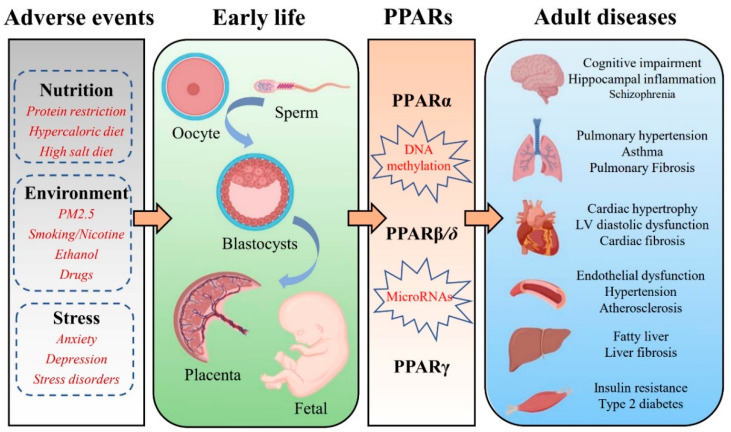
Schematic illustration of potential mechanisms to support the association between early-life adverse events exposure and epigenetic regulation of PPARs that promote chronic diseases in adulthood.

**Table 1 cells-11-03474-t001:** Summary of studies on PPARs in the fetal origins of adult disease.

Organs	Adverse Factors	PPARs	Adverse Outcomes/Phenotype	Reference
Brain	Maternal dietary restriction	PPARα	Abnormal sleep homeostatic regulation	[140]
	Maternal immune activation		Disruption of dopamine function	[141]
	Maternal vitamin D deficiency	PPARγ	Angiogenesis impairment	[142]
	Maternal immune activation		Cognitive impairments and anxiety behaviors	[143]
	Maternal high fructose		Hippocampal neuroinflammation	[144]
	Intrauterine growth restriction		Neurodevelopment and neurocognitive impairment	[145]
Lung	Perinatal nicotine exposure	PPARγ	Asthma	[146]
	Perinatal nicotine exposure		Lung dysplasia	[147]
	Perinatal nicotine exposure		Lung mitochondrial dysfunction	[148]
	Intrauterine growth restriction		Impairment of lung development	[149]
Heart	Maternal protein restriction	PPARα	Dysregulation of lipid metabolism	[150]
	Maternal diabetes		Fetal hypertrophic cardiomyopathy	[132]
	Maternal diabetes		Cardiac oxidative stress	[151]
	Maternal obesity	PPARγ	Fetal cardiac dysfunction	[152]
	Maternal protein restriction		Cardiac fibrosis	[153]
	Maternal nutrient restriction		Myocardial lipid deposition	[154]
Vessel	Preeclampsia	PPARβ/δ	Endothelial dysfunction	[155]
	Maternal protein restriction	PPARγ	Aortic remodeling	[156]
Liver	Maternal exposure to PFOA	PPARα	Liver damage	[157]
	Maternal high-fat diet		Non-alcoholic fatty liver disease	[158]
	Maternal nicotine exposure		Metabolic-associated fatty liver disease	[159]
	Unbalanced folates/vitamin B12 diet		Lipid metabolism impairment	[160]
	Maternal high-fat diet		Obesity	[161]
Liver	Maternal ethanol exposure	PPARα	Non-alcoholic fatty liver disease	[162]
	Paternal hyperglycemia		Hepatic steatosis	[163]
	Maternal high-fat feeding	PPARγ	Metabolic dysfunction	[164]
	Maternal bisphenol A exposure		Non-alcoholic fatty liver disease	[165]
Skeletal muscle	Maternal protein restriction	PPARα	Metabolic inflexibility	[166]
	Intrauterine growth retardation	PPARβ/δ	Insulin resistance	[167]
	Maternal/Paternal type 2 diabetes		Insulin resistance	[168]
	Intrauterine growth retardation	PPARγ	Insulin resistance	[169]
	Maternal cafeteria diet		Skeletal muscle development and metabolic disorders	[170]

PFOA: perfluorooctanoic acid.

**Table 2 cells-11-03474-t002:** Summary of studies on PPAR agonists in the fetal origins of adult disease.

Types	Drugs	Rescued Phenotype	Reference
PPARα agonist	Clofibrate	Fatty liver disease	[238]
	Fenofibrate	Disruption of dopamine function	[141]
	WY-14643	Obesity	[255]
PPARβ/δ agonist	GW1516	Endothelial dysfunction	[224]
PPARγ agonist	Rosiglitazone	Asthma	[256]
	Pioglitazone	Neuroinflammation and oxidative stress	[257]
	Pioglitazone	Learning and memory abilities impairment	[258]
	Rosiglitazone	Cardiac adverse remodeling	[153]
	Rosiglitazone	Skeletal muscle insulin sensitivity	[254]
	Rosiglitazone	Blood pressure and aortic structure	[156]

## Data Availability

Not applicable.

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
