# Peer review of "The Potential Role of PPARs in the Fetal Origins of Adult Disease"

_cells, 2022, doi:10.3390/cells11213474_

Round 1

Reviewer 1 Report

This is a very nice and well-written review. The authors are focusing on PPARs in the fetal origins of adult disease (FOAD) and DOHaD theory.

It is very informative and provides new perspectives of PPARs in the fetal origins of adult disease (FAOD). It is a unique review to shed light on the relationship between PPARs and FAOD. The roles of PPARs in FAOD would be very interesting topic for readers. Overall, it was a good review, but I thought that the authors need to focus on FAOS more. Current manuscript is ambiguous which FAOS or DOHaD would be focused.

There are some comments on this manuscript as below.

1) In the title, the authors described the fetal origins of adult disease (FOAD), however, the authors did not explain well about the roles of PPAR family proteins in the placenta or fetal development.

In addition, the authors also mentioned on the DOHaD theory, and it made more difficult to follow. It is helpful for readers to provide more information or figures about the roles of PPAR family proteins in the placenta or fetal development around line 89-102. The authors can be added the function of PPARs during placental expression or the following effects on epigenetic regulations (e.g. DNA methylation, histone modification, miRNAs and lncRNA etc.).

If the authors would like to focus on the fetal origins of adult disease (FOAD), please add more schemes or figures on the roles of PPARs in utero.

Also, it would be interesting if the authors summarized the current update on the DNA methylation on the PPAR family or related proteins.

2) 2. Early life adverse exposure and future disease risk (Line103-)

The authors described on the early life adverse exposures such as nutrition, environment, or stress and they summarized nicely. However, the authors only described about DOHaD, not for FOAD. I had an impression that this paragraph is written about DOHaD (I understood DOHaD theory is came from FOAD and these theories are quite similar).  If the authors would like to focus on FAOD, they need to be sticked to FAOD consistently instead of DOHaD.  

3). 5. Conclusion and Outlook (line 512-)

The authors need to be addressed on the future perspectives on this topic.

For example, Yuan et al. demonstrated that pharmacologic activation of PPARα (Wy-14643) treatment during the suckling period enhanced reduced DNA methylation on Fgf21 gene and attenuated diet-induced obesity in adulthood (Nat Commun 9, 636 (2018). https://doi.org/10.1038/s41467-018-03038-w).

As the authors suggested that the application of PPARs agonists in the primary prevention and treatment of FOAD remains controversial right now. However, it would be better to add more positive perspectives on FOAD (or DOHaD) prevention. With the future applications based on maternal treatment/supplementation with PPAR related drugs, the readers would be imagined the potential of PPARs in this specific field.

4) Figure 1 should be introduced more earlier page (e.g. Section2 early adverse events) because this figure shows the key concept on this manuscript.

Author Response

Reviewer #1

This is a very nice and well-written review. The authors are focusing on PPARs in the fetal origins of adult disease (FOAD) and DOHaD theory. It is very informative and provides new perspectives of PPARs in the fetal origins of adult disease (FAOD). It is a unique review to shed light on the relationship between PPARs and FAOD. The roles of PPARs in FAOD would be very interesting topic for readers. Overall, it was a good review, but I thought that the authors need to focus on FAOS more. Current manuscript is ambiguous which FAOS or DOHaD would be focused. There are some comments on this manuscript as below.

Dear Reviewer:

Thank you very much for your careful review and comments for helping us to improve the quality of our manuscript. We have revised the manuscript following your comments. The detailed point-by-point answers to your concerns are given below:

1). In the title, the authors described the fetal origins of adult disease (FOAD), however, the authors did not explain well about the roles of PPAR family proteins in the placenta or fetal development. In addition, the authors also mentioned on the DOHaD theory, and it made more difficult to follow. It is helpful for readers to provide more information or figures about the roles of PPAR family proteins in the placenta or fetal development around line 89-102.The authors can be added the function of PPARs during placental expression or the following effects on epigenetic regulations (e.g. DNA methylation, histone modification, miRNAs and lncRNA etc.). If the authors would like to focus on the fetal origins of adult disease (FOAD), please add more schemes or figures on the roles of PPARs in utero. Also, it would be interesting if the authors summarized the current update on the DNA methylation on the PPAR family or related proteins.

Response: We thank you for raising this important point and apologies for not making this obvious in our manuscript. To address this specific comment, we have added 2 paragraphs in Section 4 (page 7-8, line 325-367) to discuss the effects of PPARs in the placenta and the fetus, including the effects on epigenetic regulations.

2) 2. Early life adverse exposure and future disease risk (Line103-). The authors described on the early life adverse exposures such as nutrition, environment, or stress and they summarized nicely. However, the authors only described about DOHaD, not for FOAD. I had an impression that this paragraph is written about DOHaD (I understood DOHaD theory is came from FOAD and these theories are quite similar).  If the authors would like to focus on FAOD, they need to be sticked to FAOD consistently instead of DOHaD. 

Response: Thanks for your suggestion. Indeed, DOHaD began life as the “Barker hypothesis” which was foundational in the history of the field. As further studies emerged exploring more maternal pregnancy exposures and offspring outcomes, the Barker hypothesis became the Fetal Origins of Adult Disease (FOAD) hypothesis, and was extended to DOHaD in 2003 to recognise the role of exposures occurring beyond pregnancy in the postnatal developmental period. Sorry for the confusion with the terminology, and we have amended it as suggested.

3). 5. Conclusion and Outlook (line 512-). The authors need to be addressed on the future perspectives on this topic. For example, Yuan et al. demonstrated that pharmacologic activation of PPARα (Wy-14643) treatment during the suckling period enhanced reduced DNA methylation on Fgf21 gene and attenuated diet-induced obesity in adulthood (Nat Commun 9, 636 (2018). https://doi.org/10.1038/s41467-018-03038-w).As the authors suggested that the application of PPARs agonists in the primary prevention and treatment of FOAD remains controversial right now. However, it would be better to add more positive perspectives on FOAD (or DOHaD) prevention. With the future applications based on maternal treatment/supplementation with PPAR related drugs, the readers would be imagined the potential of PPARs in this specific field.

Response: Thanks for the constructive comment. This is a pretty good idea. According to your advice, we added a new supplemental Table 2 (page 14, line 642) to summarize the analysis of positive perspectives on FOAD prevention.

4) Figure 1 should be introduced more earlier page (e.g. Section2 early adverse events) because this figure shows the key concept on this manuscript.

Response: Agree. We have adjusted the position of Figure 1 to the introduction section (page 2-3, line 100-112) based on your comment.

Reviewer 2 Report

This review “The potential role of PPARs in the fetal origins of adult disease” aims at summarizing the latest research on PPAR in the field of fetal origin of adult disease theory. The manuscript is well organized and clearly written. However, except some scarse mentions of methylation, the authors do not describe the crucial role of post translational modifications on PPAR functions. Moreover, regarding methylation, the effect of vitamin B9 and B12 maternal deficiencies should be commented. In addition, the mechanisms linking PPAR to energy metabolism and mitochondrial functions are surprisingly absent regarding their preeminent roles in the pathophysiology of metabolic diseases.

Author Response

Reviewer #2

This review “The potential role of PPARs in the fetal origins of adult disease” aims at summarizing the latest research on PPAR in the field of fetal origin of adult disease theory. The manuscript is well organized and clearly written.

Dear Reviewer:

Thank you very much for your careful review and comments for helping us to improve the quality of our manuscript. We have revised the manuscript following your comments. The detailed point-by-point answers to your concerns are given below:

However, except some scarce mentions of methylation, the authors do not describe the crucial role of post translational modifications on PPAR functions.

Response: Thanks for your careful review for our manuscript. We appreciate your constructive comments. In the revised manuscript, we have added relevant content of post translational modifications on PPAR functions (page 5-7, line 212-367, highlighted in yellow).

Moreover, regarding methylation, the effect of vitamin B9 and B12 maternal deficiencies should be commented.

Response: We thank the referee for this question. In our previous manuscript, this point was indeed not discussed. Following your suggestion, we have added the requested information accordingly in the revised manuscript (page 13, line 558-561).

In addition, the mechanisms linking PPAR to energy metabolism and mitochondrial functions are surprisingly absent regarding their preeminent roles in the pathophysiology of metabolic diseases.

Response: Thanks for the question and we are sorry for the lack of description of the mechanisms linking PPAR to energy metabolism and mitochondrial functions. We have added description of some possible mechanisms in Section 3 (page 5-7, line 212-367, highlighted in yellow).

Round 2

Reviewer 2 Report

Thank you for having modified your manuscript according to previous comments. The revised version of the manuscript has been improved by these additional relevant data